# Impact of Early Mobilization on Recovery after Major Head and Neck Surgery with Free Flap Reconstruction

**DOI:** 10.3390/cancers13122852

**Published:** 2021-06-08

**Authors:** Rosie Twomey, T. Wayne Matthews, Steven Nakoneshny, Christiaan Schrag, Shamir P. Chandarana, Jennifer Matthews, David McKenzie, Robert D. Hart, Na Li, Khara M. Sauro, Joseph C. Dort

**Affiliations:** 1Ohlson Research Initiative, Arnie Charbonneau Research Institute, University of Calgary Cumming School of Medicine, 3280 Hospital Dr. NW, Calgary, AB T2N 4Z6, Canada; rosemary.twomey@ucalgary.ca (R.T.); wmatthew@ucalgary.ca (T.W.M.); scnakone@ucalgary.ca (S.N.); cschrag@ucalgary.ca (C.S.); shamir.chandarana@ucalgary.ca (S.P.C.); jennifer.matthews2@ucalgary.ca (J.M.); charles.mckenzie@albertahealthservices.ca (D.M.); robert.hart@albertahealthservices.ca (R.D.H.); 2O’Brien Institute of Public Health, Cumming School of Medicine, University of Calgary, 3280 Hospital Dr. NW, Calgary, AB T2N 4Z6, Canada; 3Section of Otolaryngology-Head & Neck Surgery, Department of Surgery, University of Calgary Cumming School of Medicine, 3280 Hospital Drive NW, Calgary, AB T2N 4Z6, Canada; 4Foothills Medical Centre, Alberta Health Services, 1403 29 St NW, Calgary, AB T2N 2T9, Canada; 5Section of Plastic and Reconstructive Surgery, Department of Surgery, University of Calgary Cumming School of Medicine, 3330 Hospital Dr. NW, Calgary, AB T2N 4N1, Canada; 6Departments of Community Health Sciences, Surgery & Oncology University of Calgary Cumming School of Medicine, 3D10, 3280 Hospital Drive NW, Calgary, AB T2N 4Z6, Canada; Na.Li@ucalgary.ca

**Keywords:** head and neck cancer, head and neck surgery, care pathways, clinical pathways, enhanced recovery, early mobilization, clinical outcomes improvement

## Abstract

**Simple Summary:**

For patients diagnosed with head and neck cancer (HNC), surgery to remove the tumour is a standard treatment. The surgery is complex-in most cases, the mouth and throat need to be rebuilt using tissue from another area of the body to restore appearance and function. Recovery from HNC surgery is challenging, and complications occur frequently. It is recommended that patients get out of bed and move (are “mobilized”) as early as possible after surgery (within 24 h) to improve recovery. However, evidence for this recommendation mainly comes from other types of cancer. Therefore, this study investigated whether early mobilization impacts recovery in patients undergoing HNC surgery. We found that delaying mobilization (after 24 h) was linked with more complications and a longer stay in the hospital. Helping patients mobilize within 24 h after HNC surgery should be a priority for healthcare teams.

**Abstract:**

Surgery with free flap reconstruction is a standard treatment for head and neck cancer (HNC). Because of the complexity of HNC surgery, recovery can be challenging, and complications are common. One of the foundations of enhanced recovery after surgery (ERAS) is early postoperative mobilization. The ERAS guidelines for HNC surgery with free flap reconstruction recommend mobilization within 24 h. This is based mainly on evidence from other surgical disciplines, and the extent to which mobilization within 24 h improves recovery after HNC surgery has not been explored. This retrospective analysis included 445 patients from the Calgary Head and Neck Enhanced Recovery Program. Mobilization after 24 h was associated with more complications of any type (OR = 1.73, 95% CI [confidence interval] = 1.16–2.57) and more major complications (OR = 1.76; 95% CI = 1.00–3.16). When accounting for patient and clinical factors, mobilization after 48 h was a significant predictor of major complications (OR = 2.61; 95% CI = 1.10–6.21) and prolonged length of stay (>10 days; OR = 2.85, 95% CI = 1.41–5.76). This comprehensive analysis of the impact of early mobilization on postoperative complications and length of stay in a large HNC cohort provides novel evidence supporting adherence to the ERAS early mobilization recommendations. Early mobilization should be a priority for patients undergoing HNC surgery with free flap reconstruction.

## 1. Introduction

Head and neck cancer (HNC) ranks amongst the top ten most common types of cancer worldwide [1]. Surgery with free flap reconstruction is a standard treatment for HNC. Resection of head and neck tumours can involve removing critical structures required for speech and swallowing, leaving patients with significant functional impairments. Patients face numerous challenges across the surgical timeline and an extended recovery in the hospital (often 10 days or longer) [2]. At least two-thirds of patients have a postoperative medical or surgical complication in the days after surgery [3]. HNC-specific care pathways optimize care before, during and after these complicated surgical procedures [2,4]. Care pathways specify perioperative interventions and timelines to achieve optimal clinical outcomes and reduce the costs of care. Enhanced recovery after surgery (ERAS) guidelines were initially developed for colorectal cancer [5] but have subsequently been developed for many other surgical disciplines [6] and have become widely adopted. An ERAS guideline for head and neck surgery with free flap reconstruction was published in 2017 [4].

One of the foundations of ERAS is early postoperative mobilization [6]. Mobilization includes sitting, standing, and walking as early as possible after surgery. Although traditionally, HNC surgeons may have recommended bed rest or activity restrictions due to concerns about vascular flap failure, these concerns remain unsubstantiated [7,8]. It is now widely recognized that early mobilization is important to optimize postoperative clinical outcomes [9]. Few studies, however, have examined the impact of early mobilization after HNC surgery with free flap reconstruction. Over the past decade, the field of enhanced recovery has advanced greatly, yet there has been only one retrospective cohort and one randomized trial investigating postoperative mobilization and clinical outcomes after HNC surgery [7,8]. Yeung et al. (2013) [7] found that patients mobilized after postoperative day (POD) four were four times more likely to develop pneumonia. Similarly, Yang et al. (2020) [8] found that an early mobilization intervention (off-bed activity on POD three) was associated with reduced time to removal of nasogastric tubes, urethral catheters and tracheostomies; and was also associated with longer sleep duration and improved patient comfort. Given the evidence in HNC and other surgical disciplines, the ERAS guideline for HNC surgery now recommends mobilization within 24 h (moderate evidence, strong recommendation) [4]. The extent to which mobilization within 24 h improves recovery has not been explored, especially in a large cohort of patients with HNC undergoing major head and neck resection and free flap reconstruction.

The Calgary Head and Neck Enhanced Recovery Program (CHERP), which has been in place since 2012, was adapted to include additional head and neck ERAS elements in late 2017, and the combined CHERP/ERAS program continues to the present time. This retrospective cohort study explored the association between early mobilization (using the CHERP/ERAS pathway) and clinical and process of care outcomes among patients with HNC undergoing surgery with free flap reconstruction. The specific objectives of this study were: (1) To describe time to postoperative mobilization and the proportion of patients mobilized within the pathway recommended timeframe; (2) to understand the association between early mobilization and postoperative complications and length of stay (LOS), including how patient and clinical variables modify the association; (3) to understand the association between patient and clinical variables and delayed postoperative mobilization.

## 2. Materials and Methods

This was a retrospective cohort study. Data were collected prospectively as standard practice in the CHERP, which is part of a comprehensive continuous quality improvement program [2,10].

### 2.1. Setting

Alberta is a province in Western Canada with a population of 4.4 million (October 2020) [11]. Healthcare in Alberta is delivered through a single health service provider (Alberta Health Services) within a universal, publicly funded healthcare system. All surgeries were performed at Foothills Medical Centre in Calgary, which is the tertiary referral centre for major HNC surgery with free flap reconstruction in Southern Alberta. The study was conducted from 4 June 2012, until 31 September 2020.

### 2.2. Participants

All patients over 18 years old who underwent resection with free flap reconstruction for head and neck malignancies, benign tumours or complications of other treatments (e.g., osteoradionecrosis) were included in this study. Patients were excluded if data for early mobilization (pathway data) were missing, if they were aged < 18 years old and if they had surgery without free flap reconstruction.

### 2.3. Enhanced Recovery Pathway

The CHERP includes an established measurement, audit and feedback system and care pathway elements are delivered using computerized order sets [2]. In addition to postoperative mobilization, the pathway includes care elements such as preadmission education, perioperative nutritional care, prophylaxis against thromboembolism, postoperative nausea/vomiting prophylaxis, urinary catheterization, tracheostomy management, postoperative flap monitoring, wound management, and pulmonary physical therapy. Two additional ERAS elements were added in December 2017: a perioperative multimodal analgesia protocol and an intraoperative fluid management protocol. The protocol for early mobilization was the same in both the CHERP and ERAS pathways. The majority of patients undergoing major resection with free flap reconstruction are managed overnight in the intensive care unit (ICU) and then transferred to a ward staffed by clinicians specialized in managing head and neck surgical patients. A minority are transferred directly to the ward.

### 2.4. Data Sources

The pathway is fully integrated into the inpatient electronic medical record, which is the source of the clinical data used in this study. Clinical data, as defined by the minimum data set, are prospectively collected by a trained research assistant who is embedded in the hospital inpatient unit [10]. Clinical data are obtained through a combination of paper chart review and data abstraction from the EMR. Information on complications is recorded in the chart by the most responsible physician (attending or resident). Charts are reviewed by a trained professional for sentinel events, and physician and allied health professional notes are reviewed to obtain information on important pathway milestones, including mobilization time. Diagnostic imaging reports are also reviewed to supplement paper charts and EMR data.

### 2.5. Bias

Data were collected prospectively by a trained professional and not by the research team, minimizing selection bias and improving reliability. We inspected the demographic characteristics of patients with and without mobilization data to check for evidence of selection bias.

### 2.6. Outcomes

To understand the association between early mobilization and postoperative complications and LOS, the primary outcomes were any complications, severe complications and a prolonged LOS. Two dichotomized variables were created to examine postoperative complications: (1) Occurrence of any complication (yes, no) was defined as any deviation from the normal postoperative course and was classified as grade I–V using the Clavien-Dindo classification [12]. (2) Major complications (yes, no) were defined as grade IIIb–V using the Clavien-Dindo classification, which includes complications requiring surgical, endoscopic or radiological intervention under general anesthesia, life-threatening complications and death [12]. In separate analyses, we also used dichotomous complication variables (yes, no) to examine specific complication types, including: pneumonia, pulmonary embolism, deep vein thrombosis, delirium tremens, myocardial infarction, bleed or hematoma, free flap compromise and failure. LOS was dichotomized as POD 0-10 and POD >10 (the latter representing a prolonged LOS), and this cut-point was based on both the median LOS and the pathway recommended LOS. In addition, LOS is also reported as a continuous variable.

### 2.7. Time to Postoperative Mobilization (Exposure)

The date of first meaningful mobilization was recorded as the date where there was evidence that the patient was mobilized out of bed, up in a chair, standing and/or walking. Time to mobilization was calculated as the number of calendar days from the date of surgery (POD 0) to the date of first meaningful mobilization. Postoperative early mobilization was also categorized as POD 0–1 (a surrogate for mobilization within 24 h postoperative, in line with current pathway recommendations [4]) vs. POD > 1 (mobilization after 24 h postoperative. Postoperative mobilization was also categorized as POD 0–2 (a surrogate for within 48 h) and POD > 2 (mobilization after 48 h postoperative). To examine the association between early mobilization and postoperative complications and LOS, delayed postoperative mobilization (after 24 h) was considered an exposure. To understand the association between patient and clinical variables and delayed postoperative mobilization, delayed postoperative mobilization was the outcome.

### 2.8. Variables

Potential predictor variables of complications, a prolonged LOS and delayed mobilization, were identified through the existing evidence and expert consultation with the research team. Preoperative variables included age (centred at the mean), sex (male or female), smoking status (never, former or current smoker), alcohol habits (never, former, light/moderate, heavy), body mass index (BMI; standard categories for underweight, healthy range, overweight and obese based on kg/m [2]), primary cancer site (oral cavity, pharynx and larynx, skin, other), cancer stage (0–II and III–IV), and the number of comorbidities (none, one or at least two). Intraoperative and postoperative variables included presence of tracheostomy, number of free flaps, flap donor type (radial forearm, fibula, anterolateral thigh, other), resection extent (soft tissue, bone or soft tissue and bone) and arrival to the hospital unit from the ICU as POD 0–1 vs. POD > 1 (a surrogate for the duration of the ICU admission).

### 2.9. Statistical Methods

All data were analyzed using Stata 16.1 (Stata Corp, College Station, TX, USA) [13] and alpha was set at 0.05 for all statistical tests. To describe patient characteristics, time to postoperative mobilization and the proportion of patients mobilized within the pathway recommended timeframe; data were summarized as frequencies (percentages) for categorical variables, range, median (interquartile range) or mean (standard deviation, SD) for continuous variables. To compare demographic and clinical characteristics between patients who were mobilized early and those who were not, two-tailed Fisher’s exact tests for r × c contingency tables (alcohol status, smoking status, comorbidities, primary site, clinical-stage, number of free flaps) and independent *t*-test (age on the date of surgery) were used [14]. To test the univariable associations between time to mobilization and postoperative complications and LOS, a two-tailed Fisher’s exact tests (any complication, major complications including pneumonia, pulmonary embolism, deep vein thrombosis, delirium tremens, myocardial infarction, bleed or hematoma, free flap compromise, flap failure, LOS dichotomized at POD 10) and a Mann U Whitney test (LOS as a continuous variable) were used.

To explore the associations between potential predictors and postoperative complications and a prolonged LOS, multivariable logistic regression was used. To improve the predictive performance of the regression models, we used a method of variable selection called elastic net regularization [15]. This method was chosen because of the high number of potential predictor variables and its ability to guide the selection of the most relevant variables while simultaneously minimizing model overfitting. This method combines the Least Absolute Shrinkage and Selection Operator (LASSO) and ridge methods of variable selection. LASSO uses a penalty function to shrink regression coefficients and selects for only the most important regressors. The elastic net extends the LASSO by adding a quadratic term to the penalty function and was designed to produce better predictions and model performance when covariates are highly correlated. We also examined the assumption of collinearity of (non-binary) predictor variables using the “collin” package in Stata and a variance inflation factor (a measure of the amount of multicollinearity in a set of multiple regression variables) cut-point of <5. Variables selected by elastic net regularization were subsequently entered into the logistical regression models. This process was repeated to examine the association between patient and clinical variables and delayed postoperative mobilization. The Akaike information criterion (AIC) is reported for each model [16]. The reference levels for the categorical variables in all models were: BMI = healthy range; sex = male; smoking status = never; alcohol status = never; comorbidity = none; primary site = Oral Cavity; Cancer stage = I–II; flap count = one; resection extent = soft tissue only; flap donor site = radial forearm; mobilization = POD 0–1 (within 24 h) and POD 0–2 (within 48 h).

## 3. Results

Mobilization (pathway) data were missing in 11% (*n* = 55) of cases. Less than 5% of data for complications and LOS were missing. We did not identify any imbalance of prognostic factors between eligible patients with and without pathway data (Appendix A), suggesting no selective exclusion of patients. Missing data were missing at random either due to failure to flag a case for data collection or a failure to record mobilization milestones in the chart.

Major resection with free flap volumes varied between a minimum of 43 and a maximum of 75 cases per year during this study. There were 500 eligible patients between 4 June 2012, and 31 September 2020. Of this sample, *n* = 445 patients had pathway data and were included in this analysis. Characteristics of the study cohort are presented in Table 1; briefly, patients were aged 61.2 ± 12.2 (mean ± SD) years, predominantly male (68%) with stage III–IV (65%) squamous cell carcinoma (80%). Postoperative mobilization time ranged from 0–32 days with a median of 2 days (IQR [interquartile range] = 1–2). We found that 44% of patients were mobilized within 24 h, 77% of patients were mobilized within 48 h, and 91% of patients were mobilized within 72 h after surgery. A minority of patients (5%) had not been mobilized by POD 4.

### 3.1. Postoperative Complications

A total of 242 (45%) of patients had a complication, and 70 (16%) of these were major, including loss of flap in five cases (1%). In univariable analyses, mobilization after 24 h was associated with more complications (OR = 1.73 [95% CI = 1.16–2.57]; *p* = 0.005) and more major complications (OR = 1.76 [95% CI = 1.00–3.16]; *p* = 0.049; Table 2). This association persisted among those mobilized after 48 h (OR for any complication = 2.08 [95% CI = 1.28–3.43]; *p* = 0.002 and OR for major complications = 2.28 [95% CI = 1.26–4.05]; *p* = 0.005). Furthermore, mobilization after 48 h was associated with more cases of pneumonia (OR = 4.07 [95% CI = 1.92–8.59]; *p* < 0.001); 5% of patients mobilized within 48 h developed pneumonia compared to 18% of patients mobilized after 48 h (Table 2 and Appendix A).

In multivariable analyses, we found that patients who were older, had one or more comorbidities and had more advanced stages of cancer were more likely to have any complication (AIC = 483; Appendix A). Patients who were current smokers had two or more comorbidities, whose flap type was “other”, and who were mobilized after 48 h were more likely to have a major complication (AIC = 312; Figure 1).

### 3.2. Hospital Length of Stay

LOS ranged from 4–114 days with an overall median of 11 days (IQR = 8–15). In univariable analyses, patients with delayed mobilization (after 24 h) were more likely to be discharged after 10 days postoperatively (OR = 1.80 [95% CI = 1.21–2.68]; *p* < 0.003). Patients who were mobilized after 48 h stayed in the hospital four days longer (median difference) than those who were mobilized within 48 h (*p* < 0.001; Table 2). In multivariable analyses, mobilization after POD 2 was a predictor of a prolonged LOS (OR = 2.85 [95% CI = 1.41–5.76]; *p* = 0.004). Other significant predictors of a prolonged LOS included older age, a tracheostomy, any complication and primary cancer site (pharynx/larynx; AIC = 384; Appendix A).

### 3.3. Delayed Mobilization

Multivariable predictors of delayed mobilization (mobilization after 24 h) were the presence of a tracheostomy (OR = 2.81 [95% CI = 1.70–6.37]; *p* < 0.001) and a longer ICU admission (OR = 3.29 [95% CI = 1.70–6.37]; *p* < 0.001; AIC = 576; Appendix A).

## 4. Discussion

In this study, we found that patients who were mobilized early had fewer complications overall, fewer major complications, and a shorter LOS. When accounting for relevant patient and clinical variables, mobilization after 48 h predicted major complications and a prolonged LOS (hospital discharge after POD 10). The main predictors of delayed mobilization were an extended ICU admission and having a tracheostomy. This comprehensive analysis demonstrates the impact of early mobilization on postoperative complications and LOS in this important surgical group. Our study provides new HNC-specific evidence to support the ERAS recommendation that patients should be mobilized within 24 h after surgery where possible.

Postoperative complications are common for patients undergoing major HNC surgery (45% in the present study) and have been associated with increased LOS, morbidity and decreased overall survival [17]. Early postoperative mobilization is an established component of ERAS based on counteracting catabolic changes with bed rest, prophylaxis of pulmonary embolism and deep vein thrombosis, and supporting a return to normal function. ERAS guidelines for HNC surgery give early mobilization a strong recommendation [4] despite relatively little evidence in this patient group. To our knowledge, an association between mobilization with 24 h, complications and LOS has not previously been reported in this patient population. In a cohort of 62 patients, our group previously reported that mobilization after POD 4 was associated with an increased incidence of pneumonia [7]. Our current study-conducted in a much larger cohort and using methods that minimize statistical modelling error-enables a more comprehensive understanding of the association between mobilization and important clinical and process outcomes. We replicate our previous finding for pneumonia and show that the incidence of pneumonia was three times higher in patients who were mobilized after POD 2 in the current cohort. Therefore, these findings suggest that ERAS recommendations pertaining to early mobilization are worthy of focused attention by clinical teams working with this patient population. Other surgical disciplines with ERAS guidelines also have explored whether early mobilization predicts objective markers of recovery, considering ERAS involves numerous care elements (17 in HNC [4]). In particular, early mobilization is a predictor of morbidity, complication severity and LOS after colorectal cancer surgery [18,19,20].

Using data from a large cohort of 445 patients collected over the past eight years, we found that the overall proportion of patients mobilized within 24 h was 44%, and 77% were moving by POD 2. Our study extends the work of others on compliance with ERAS early mobilization targets after HNC surgery [21,22,23,24,25]. In 2016, motivated by data showing suboptimal mobilization performance, our pathway target was adjusted with beneficial results. These efforts to improve mobilization compliance since 2016 are described in our companion paper. To improve mobilization, barriers to mobilization compliance were explored. We found that primary predictors of delayed mobilization are a longer ICU admission and the presence of a tracheostomy. To facilitate early mobilization, other institutions with ERAS protocols avoid ICU admission if medically suitable, through a combination of a rapid wake-up protocol developed by the anesthesia team and tracheostomy care and free flap monitoring taking place on the unit by a trained nursing team [21]. Where feasible, this requires further consideration in future advancements of HNC-specific ERAS pathways. However, we believe that future research should also explore the benefits of physical activity in the waiting period before HNC surgery [26] and build on promising early evidence that postoperative exercise (in addition to standard physical therapy) reduces LOS [27].

A limitation of the present analysis and the wider body of research on early mobilization after surgery lies in the accuracy of the measurement of mobilization. In the CHERP and many other enhanced recovery pathways for HNC surgery, mobilization has only been evaluated as the interval of days between surgery and mobilization, with variable definitions of what counts as mobilization [28]. The need for consistent charting places an undue burden on the healthcare team [29], and although this metric indicates how quickly a person begins to mobilize after surgery, it provides no information on how much. Avoiding bed rest, preventing functional decline and progressing towards preoperative activity levels helps patients meet discharge criteria earlier and sets the stage for a seamless transition to recovery at home. ERAS pathways for HNC surgery have begun to measure step counts objectively using pedometers [25], and commercially-available wearable technology offers a more sophisticated solution that could be integrated within the EMR. An objective measure of mobilization would improve understanding of the dose-response relationship: the quantity of mobilization (e.g., number of daily step counts) that is safe and associated with improved patient outcomes [30]. Wearable technology can provide continuous feedback on activity across the hospital stay, which can be used by patients, the clinical team, and in prospective trials involving mobilization interventions [29].

Our study has several strengths, including a large cohort of consecutively treated patients (which permits exploration of more covariates and enables better statistical modelling), treated with a consistent care team and computerized order entry (which facilitates more reliable delivery of clinical care). However, there are some limitations that should be considered when interpreting our findings. Data were collected within one centre in Alberta, Canada, and we acknowledge that enhanced recovery pathways must be tailored to the local setting and resources. Although we have extended the evidence of the importance of mobilization within 24 h after HNC surgery, similar analyses from other institutions will support the generalizability of our findings. The retrospective nature of this study limited our ability to assess the timing of all complications in relation to early mobilization, limiting our understanding about the causal relationship between mobilization and complications. Some of the complications that are most likely to occur early in the postoperative period are bleeding, hematoma, and flap compromise [31,32]. However, in the majority of cases, patients were mobilized before these major complications are likely to have occurred. Therefore, we do not believe that the lack of high-resolution data on the timing of complications confuses the associations described in this study. Similarly, data on postoperative nausea and vomiting was not extractable for data analysis, though adequate control of this symptom is needed for effective mobilization [33]. Finally, we were unable to include the Charlson comorbidity index. Establishing causality requires other research designs, and the results of this study should be interpreted with due caution. However, our data add to findings from other surgical specialties and suggest that interventions that target mobilization during the perioperative period for patients undergoing HNC surgery are an avenue for future research to improve clinical outcomes.

## 
5. Conclusions


Mobilization within 24 h was associated with fewer complications, fewer major complications, and a significantly shorter LOS in a large sample of patients undergoing head and neck surgery with free flap reconstruction. Mobilization after 48 h was a predictor of major complications and a longer LOS. Therefore, early mobilization should be a priority when caring for patients undergoing major HNC resection with free flap reconstruction and should be included in enhanced recovery pathways as recommended by ERAS guidelines. Future prospective research integrating wearable technology may provide further insight into the relationship between increased mobilization and improved patient outcomes.

## Figures and Tables

**Figure 1 cancers-13-02852-f001:**
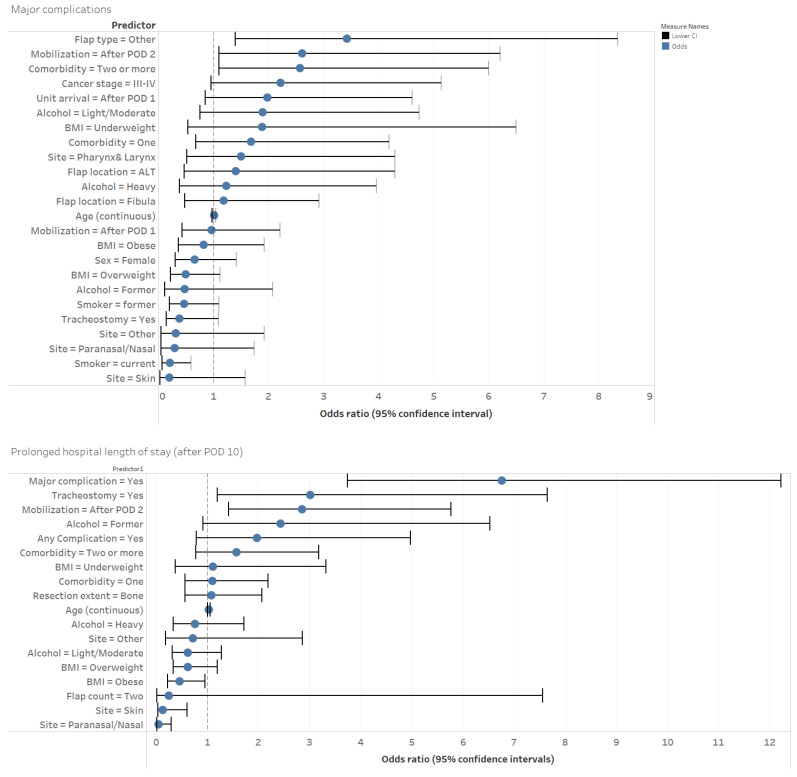
A visual representation of the odds ratios and 95% confidence intervals for the multivariable logistic regression models of major postoperative complications (top panel) and a prolonged LOS (bottom panel). BMI = body mass index. POD = postoperative date. Flap count and resection extent were removed from this visualization due to wide confidence intervals (Appendix A). The reference levels for the categorical variables were: Alcohol status = never; BMI = healthy range; sex = male; smoking status = never; alcohol status = never; comorbidity = none; primary site = Oral Cavity; Cancer stage = I–II; flap count = one; resection extent = soft tissue only; flap donor site = radial forearm; mobilization = POD 0–1 (within 24 h) and POD 0–2 (within 48 h).

**Table 1 cancers-13-02852-t001:** Patient characteristics in for the cohort and for patients mobilized within (POD 0-1) or after (POD > 1) 24 h.

Characteristic	Cohort *n* (%) *n* = 445	POD 0–1 *n* (%) *n* = 196	POD >1 *n* (%) *n* = 249	*p*-Value
**Sex**				
Male	303 (68)	132 (67)	171 (69)	0.838
Female	142 (32)	64 (33)	78 (31)	
**Age (years)**				
Mean ± SD	61.2 ± 12.2	61.4 ± 12.7	61.0 ± 12.2	0.371 *
Range	21.2–89.0			
**Alcohol status**				0.810
Never	90 (20)	36 (21)	60 (23)	
Light/Moderate	162 (36)	75 (43)	102 (38)	
Heavy	93 (21)	41 (24)	73 (27)	
Former	48 (11)	22 (13)	32 (12)	
Not reported	52 (12)			
**Smoking status**				0.664
Never smoked	117 (26)	52 (29)	65 (28)	
Former smoker	151 (34)	62 (35)	89 (39)	
Current smoker	136 (31)	63 (36)	73 (32)	
Not reported	41 (9)			
**Comorbidities**				0.660
None	142 (32)	64 (33)	78 (31)	
One	136 (31)	63 (32)	73 (29)	
Two or more	167 (37)	69 (35)	98 (39)	
**Specific Comorbidity**				
Diabetes	54 (12)	25 (13)	29 (12)	0.771
COPD	50 (11)	22 (11)	28 (11)	1.000
Hypertension	181 (41)	70 (36)	111 (45)	0.065
Heart disease	59 (13)	22 (11)	37 (15)	0.324
**Primary site**				0.011
Oral cavity	303 (68)	121 (62)	182 (73)	
Pharynx & Larynx	42 (8)	18 (9)	24 (10)	
Skin	39 (9)	27 (14)	12 (5)	
Paranasal/Nasal	27 (6)	14 (7)	13 (5)	
Other	34 (8)	16 (8)	18 (7)	
**Histology**				0.098
Squamous cell	356 (80)	150 (77)	206 (84)	
Other cancer	83 (18)	38 (20)	38 (16)	
Benign	6 (1)	5 (3)	1 (0)	
Not reported	7 (2)			
**Clinical stage**				0.140
0	8 (2)	1 (1)	7 (3)	
I	44 (10)	21 (12)	23 (10)	
II	70 (16)	38 (21)	32 (14)	
III	66 (15)	29 (16)	37 (16)	
IV	223 (50)	88 (50)	135 (57)	
Not reported	34 (8)			
**Number of free flaps**				0.828
One	423 (95)	187 (95)	236 (95)	
Two	22 (5)	9 (5)	13 (5)	
**Flap type**				0.359
Radial forearm	235 (53)	101 (52)	134 (54)	
Fibula	95 (21)	37 (19)	58 (23)	
Anterolateral thigh	57 (13)	30 (15)	27 (11)	
Other	58 (13)	28 (14)	30 (12)	
**Resection Extent**				0.267
Soft tissue	329 (74)	152 (78)	177 (71)	
Bone	97 (22)	38 (19)	59 (24)	
Soft tissue & bone	19 (4)	6 (3)	13 (5)	

Fisher’s exact tests were used for *p*-values, except for age (* independent *t*-test).

**Table 2 cancers-13-02852-t002:** Postoperative mobilization time, complications, and length of stay. The reference level for odds ratios is indicated with *italics*.

Characteristic	All Cases	POD 0–1	POD > 1	OR	95% CI	*p*-Value	POD 0–2	POD > 2	OR	95% CI	*p*-Value
*n* (%)	445 (100)	196 (44)	249 (56)				342 (77)	103 (23)			
**Any complication**											
Yes	242 (45)	92 (47)	150 (60)	1.73	1.16–2.57	0.005	172 (50)	70 (68)	2.08	1.28–3.43	0.002
*No*	202 (55)	104 (53)	98 (40)				169 (50)	33 (32)			
**Major complication**											
Yes	70 (16)	23 (12)	47 (19)	1.76	1.00–3.16	0.049	44 (13)	26 (25)	2.28	1.26–4.05	0.005
*No*	374 (84)	173 (88)	201 (81)				297 (87)	77 (75)			
**Pneumonia**											
Yes	37 (8)	11 (6)	26 (10)	1.90	0.91–4.51	0.083	18 (5)	19 (18)	4.07	1.92–8.59	<0.001
*No*	408 (92)	185 (94)	223 (90)				324 (95)	84 (82)			
**Pulmonary embolism**											
Yes	2 (0)	0 (0)	2 (1)	-	-	0.506	2 (1)	0 (0)	0	0–6.41	1.000
No	443 (100)	196 (100)	247 (99)				340 (99)	103 (100)			
**Delirium tremens**											
Yes	19 (4)	2 (1)	17 (7)	7.11	1.64–63.98	0.003	5 (1)	14 (14)	10.60	3.46–38.36	<0.001
*No*	426 (96)	194 (99)	232 (93)				337 (99)	89 (86)			
**Deep vein thrombosis**											
Yes	3 (1)	0 (0)	3 (1)	-	-	0.259	2 (1)	1 (1)	1.67	0.03–32.28	0.547
*No*	442 (99)	196 (100)	246 (99)				340 (99)	102 (99)			
**Bleeding**											
Yes	18 (4)	6 (3)	12 (5)	1.60	0.54–5.30	0.469	12 (4)	6 (6)	1.70	0.51–5.04	0.389
*No*	427 (96)	190 (97)	237 (95)				330 (96)	97 (94)			
**Myocardial infarction**											
Yes	7 (2)	1 (1)	6 (2)	4.81	0.57–222.55	0.141	3 (1)	4 (4)	4.57	0.75–31.55	0.053
*No*	438 (98)	195 (99)	243 (98)				339 (99)	99 (96)			
**Flap compromise**											
Yes	38 (9)	16 (8)	22 (9)	1.09	0.53–2.29	0.865	27 (8)	11 (11)	1.39	0.60–3.04	0.421
*No*	407 (91)	180 (92)	227 (91)				315 (92)	92 (89)			
**Flap loss**											
Yes	5 (1)	2 (1)	3 (1)	1.18	0.13–14.29	1.000	2 (1)	3 (3)	5.1	0.57–61.55	0.084
*No*	440 (99)	194 (99)	246 (99)				340 (99)	100 (97)			
**OR return**											
Yes	68 (15)	24 (12)	44 (18)	1.54	0.87–2.75	0.1440	46 (13)	22 (21)	1.75	0.94–3.16	0.061
*No*	377 (85)	172 (88)	205 (82)				296 (87)	81 (79)			
**Length of stay**											
*POD 0-10*	204 (46)	106 (54)	98 (40)	1.80	1.21–2.68	0.003	178 (52)	26 (25)	3.23	1.93–5.51	<0.001
After POD 10	240 (55)	90 (46)	150 (60)				163 (48)	77 (75)			
Median (IQR) *	11 (8–15)	10 (8–14)	11 (9–16)	1 **	−0.1–2.1	0.003	10 (8–14)	14 (10–19)	4 **	2.6–5.4	<0.001

* Mann Whitney Test (all other tests are Fisher’s exact). ** Median difference. CI, confidence interval; OR, operating room; POD, postoperative day.

## Data Availability

The data for this study are under the custodianship of Alberta Health Services (AHS) and are therefore unavailable for sharing. Data can be made available after an appropriate data sharing, and access agreement is formally completed. Please contact Dort for more information (jdort@ucalgary.ca).

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
