# Peer review of "Impact of Early Mobilization on Recovery after Major Head and Neck Surgery with Free Flap Reconstruction"

_cancers, 2021, doi:10.3390/cancers13122852_

Round 1
Reviewer 1 Report
This is a retrospective study on 445 patients treated with extensive head and neck procedures. It is demonstrated that early mobilization as recommended by recently published ERAS guidelines is beneficial for the patients, since associated with a reduction of major complications.
It is shown that major factors influencing mobilization are tracheostomies and longer ICU admissions. Major complications significantly increasing with late mobilizations are for the most part pneumonias.
This is an extremely important finding and will guide post-operative management of these head and neck cancer patients in the future.
Should be accepted as is.
Reviewer 2 Report
Dear Authors,
thank you for submiting the manuscript. It is very well written, with clearly stated purposes,methods and results description.
However, the similarity to manuscript cancers-1213921 is huge in context of aim, methods and results. My recommendation is to combine the two manuscripts and resubmit as one.
Reviewer 3 Report
This is an interesting study about the impact of early mobilization on recovery after major head and neck surgery with free flap reconstruction. The authors analyzed 445 patients. Thay found that mobilization after 24 hours was associated with more complications.
The paper is well written. However, some issues remain.
I think that wound infection represent an important complication of head and neck surgery and must be included in the analyses.
Table 1 must report also types of demolitive surgery before flap reconstruction.
Time from surgery to complication is important to assess the role of postoperative mobilization as a cause or an effect. Therefore it must be added and included in the analyses. Moreover, this concept should be discussed in order to identify which variable is the true predictor of the other.
I think that supplementary table S3 should be included in the paper because it reports data about every single complication.
Round 2
Reviewer 2 Report
Dear Authors,
thank you for the explanation, I have no further comments.
Reviewer 3 Report
Thank you for improving the paper.